# A Novel Low-Power Synchronous Preamble Data Line Chip Design for Oscillator Control Interface

**Shih-Lun Chen [1],\*** , **Tsun-Kuang Chi [1]** , **Min-Chun Tuan [1]** , **Chiung-An Chen [2],\*** , **Liang-Hung Wang [3]** , **Wei-Yuan Chiang [2],\*** , **Ming-Yi Lin [4] and Patricia Angela R. Abu [5]**

1.  Department of Electronic Engineering, Chung Yuan Christian University, Chung Li City 320, Taiwan; balance578942@gmail.com (T.-K.C.); a25097129@hotmail.com (M.-C.T.)
2.  Department of Electrical Engineering, Ming Chi University of Technology, New Taipei City 243303, Taiwan
3.  Department of Microelectronics, College of Physics and Information Engineering, Fuzhou University, Fuzhou 350108, China; eetommy@fzu.edu.cn
4.  Department of Electrical Engineering, National United University, Miaoli 36003, Taiwan; mylin@nuu.edu.tw
5.  Department of Information Systems and Computer Science, Ateneo de Manila University, Quezon City 1108, Philippines; pabu@ateneo.edu
*   Correspondence: chrischen@cycu.edu.tw (S.-L.C.); joannechen@mail.mcut.edu.tw (C.-A.C.); jameschiang@mail.mcut.edu.tw (W.-Y.C.); Tel.: +886-3-265-4602(ext. 4610) (S.-L.C.); +886-2-2908-9899 (ext. 4813) (C.-A.C.); +886-2-2908-9899 (ext. 4827) (W.-Y.C.)

**Abstract:** In this paper, a novel low-power synchronous preamble data line protocol chip design for serial communication is proposed. The serial communication only uses two wires, chip select (CS) and secure digital (SD), to transmit and receive data between two devices. The proposed protocol aims to use a fewer number of wires for the interface, therefore reducing the complexity as well as the area of the chip design. Moreover, it increases the efficiency through a synchronous serial communication-controlled oscillator. The low-power synchronous preamble data line protocol design was successfully verified using a field-programmable gate array (FPGA) as a master device and a real chip as a slave device. The signals are checked through the use of a logic analyzer. The realized low-power synchronous preamble data line protocol chip design has a gate count of only 5.07 K gates, a low power dissipation of 12 mW, and a chip area of 453,260 μm$^2$ using the Taiwan semiconductor manufacturing company (TSMC) 0.18 μm CMOS process. Compared with the three-wire serial peripheral interface (SPI) protocol, the proposed design has the advantages of having a lower cost and a lower power consumption.

**Keywords:** SPI; digital signal process; communication protocols; CMOS digital integrated circuit; field-programmable gate array (FPGA); electronic device measurement and very-large-scale integration (VLSI)

## 1. Introduction

Moore's Law expresses that the number of transistors increases every year, leading to microelectronic devices becoming more and more common these days. From a personal computer to a mobile phone, each person has at least a microelectronic device in hand. After the mobile phone generation, wireless body sensor networks (WBSNs) [1] and the Internet of Things (IoT) [2–4] were implemented in both the medical and industrial field with radio frequency identification (RFID) techniques. The use of microelectronic devices has greatly increased in other fields of application. For each microelectronic device, timing is the most important parameter in order to keep devices working and communicate with other devices. The oscillator circuit is designed to produce a periodic analog signal as a reference for the trigger and timing effect consideration.

Generally speaking, the oscillator circuit is designed with an adjusted controller for tuning different operating frequencies to cover various kinds of applications. For analog circuits, Ref. [5] provided a good method for a lower power consumption and a wide tuning voltage controller oscillator (VCO) without inter buffer stages, while [6] proposed a low-cost VCO which also maintained the phase noise property. Both previous studies, Refs. [5,6] were realized using 0.18 µm and 0.13 µm CMOS processes, respectively. With the application of a mobile phone, Ref. [7] offered a good explication to combine a voltage controller and a temperature controller together, which is called VC–TCXO, and applying it on a driver chip. Its schematic kept the radio frequency (RF) functional and was realized on a silicon-germanium (SiGe) heterojunction bipolar transistor (HBT) process. Although the researchers provided outstanding contributions for voltage and temperature controllers, the stability and precision of tuning the frequency remained a challenge for the controllers in analog oscillators.

Therefore, Refs. [8–10] proposed a controller using a digital control bit to represent the different operating frequencies and output the corresponding frequency. Moreover, the controlling mechanism was smart and adaptive. Thus, it was easier to realize and distinguish the output frequency on a CMOS process. With that, the microelectronic device became precise and complex. For the advanced functions that are required in the controller, the method of digital control bits or circuits is not enough for communicating between a device or chip to implement. There is a need for an efficient way to connect two different devices or ICs. With that, the chip-to-chip communication protocol was released and a micro control unit (MCU) was proposed for it to work effectively. In the MCU application, Chen et al. [11] proposed to implement an MCU communication with analog to digital convertor (ADC) and RF circuits. An asynchronous interface was used to connect the different time-domain circuits by a handshake circuit. This was successfully realized using the TSMC 0.13 µm CMOS process that is low cost and with a high performance. For power consumption consideration, [12] provided a new MCU architecture for the wireless body sensor network (WBSN) application. The MCU successfully combined the filters, an encryption module, and an error correction coding (ECC) module. Its data were transmitted through a universal asynchronous receiver/transmitter (UART) interface. In its schematic design, the cost and power consumption are important parameters that can affect the system or device performance.

Serial communication is widely used in microelectronic device networks to maintain the system cost and performance. Compared to parallel communication, serial communication uses a fewer number of ports to work that provides it with its lower cost advantage. It is a strong benefit for system or IC design. Since Motorola Inc. proposed the serial peripheral interface (SPI) protocol [13] in 2003, SPI has become commonly used in microelectronic devices, embedded systems and very-large-scale integration (VLSI) communication. On the other hand, $I^2C$ was also a good alternative as a communicating means between ICs. Leens [14] provided an excellent explanation about the comparison of SPI and $I^2C$. SPI is a good choice for building a custom platform due to its flexibility, expandability and variability. With regard to transmission speed, SPI with full-duplex function can go over 10 Mb/s but $I^2C$ is only limited to 3.4 Mb/s, without full-duplex. Until now, SPI is a better option for serial communication. However, the bus size of an SPI protocol remains to be a challenge to keep the chip cost low. In SPI implementation, there is at least four signals in order to build the SPI network. However, $I^2C$ only needs two lines for building the whole $I^2C$ network.

To avoid transmission error, [15,16] provided a data error-correcting method to solve the problem. An error correcting code was realized on a Spartan 6 field-programmable gate array (FPGA) as a board with low complexity in [15]. A low-cost chip design includes a data detecting, correcting, encrypting and decrypting which was implemented using TSMC 0.18 µm process in [16]. The previous studies proposed good solutions for transmission error with low-cost or low-complexity chip design.

Consequently, it is necessary to develop a new approach to maintain the benefits of speed and overcome the bus problem as well. The study in [17] presented a three-wire SPI protocol and implemented it using a TSMC 0.18 µm CMOS process with low power and low chip area. With the bidirectional data channel technique, the proposed low power and low chip area three-wire SPI

design was successfully verified on an application-specific integrated circuit (ASIC) system and on a field-programmable gate array (FPGA).

Given the previous works and current challenges on SPI protocol, this study was devoted to developing a new approach to improve the SPI bus problem that is used as an oscillator controller in most microelectronic devices. Therefore, this paper proposes a novel low-power synchronous preamble data line protocol interface for the oscillator control application that uses a synchronous technique to reduce the gate count instead of the asynchronous interface. For verification, [17–23] provided efficient ways to design and test an SPI protocol while [17,21–23] showed that an FPGA is an effective way to implement the verification environment. The proposed protocol is realized to a circuit design using the TSMC 0.18 μm CMOS process and was successfully verified on FPGA and ASIC systems.

## 2. Overview of a Four-Wire and a Three-Wire SPI Protocol

Serial peripheral interface (SPI) protocol is a very common communication between master devices and slave devices, such as a secure digital (SD) card. The difference between transmission is that the parallel or serial types of data use two data lines instead of more than three data lines. The reduced number of lines also decreases the number of pins in the hardware therefore lowering the cost of the system. With these benefits, the system on chip (SoC) also includes the SPI protocol interface to communicate with other devices even more since the multiple sets of SPI interface are applied. In general, SPI protocol is designed with four lines to connect different devices which are SCLK (Serial Clock), SS (Slave Select), MOSI (Master Output, Slave Input), and MISO (Master Input, Slave Output) as illustrated in Figure 1.

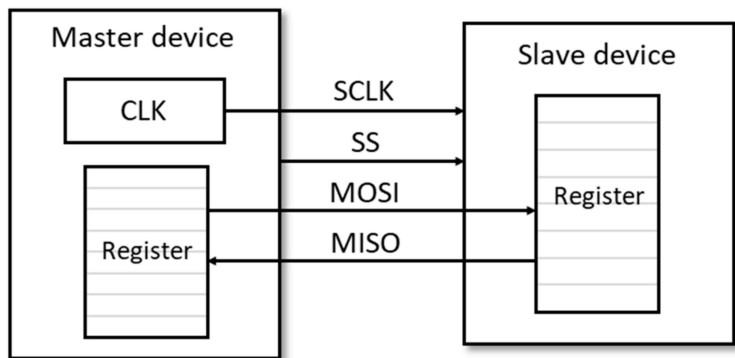

**Figure 1.** Traditional 4-wire serial peripheral interface (SPI) protocol block diagram.

The communication in a one-slave model has only one line for slave selection. However, for multiple slave devices, there are two kinds of methods to realize the operation. Figure 2a shows a multiple slave selection lines to support different slave devices. Each slave device communicates individually when the corresponding SS signal acted low. For example, there are three slave devices in the transmission system, the SS signal lines can be designed using three separate lines, SS1, SS2, and SS3 where the master device is able to communicate with the three slave devices, respectively. Another configuration is the daisy chain as shown in Figure 2b. All of the slave devices are chained together. In configuring these multiple slave devices, SPI protocol implementation only requires one SS signal line instead of a separate SS line for connecting each slave device. In some chip datasheets, SS is named CS to express chip selection and with the same meaning as slave selection.

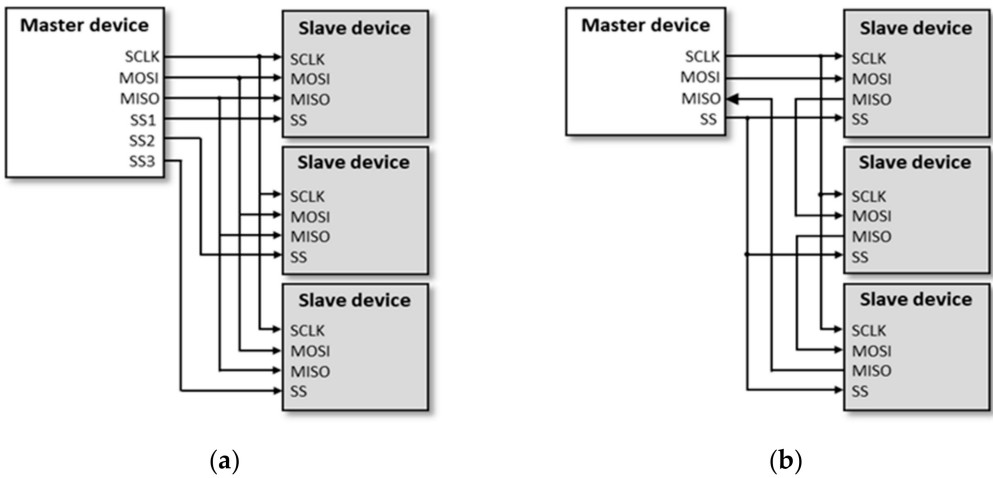

**Figure 2.** Multiple slave devices configuration diagram: (**a**) independent slave device configuration of SPI protocol block diagram and, (**b**) daisy chain slave device configuration of SPI protocol block diagram.

For customization, the master device can configure the clock polarity (CPOL) and the clock phase (CPHA) to choose the correct timing to get data in communication between the slave devices. Clock polarity (CPOL) decides the clock voltage level when communication is idle; when CPOL = 0, the clock is low level; and when CPOL = 1, the clock is high level. Clock phase (CPHA) refers to which clock edge will receive data. When CPHA = 0, the device will receive data at the first clock edge; and when CPHA = 1, the device will receive data at the second clock edge. These two configuration parameters determine the four situations for SPI communication as illustrated in Figure 3.

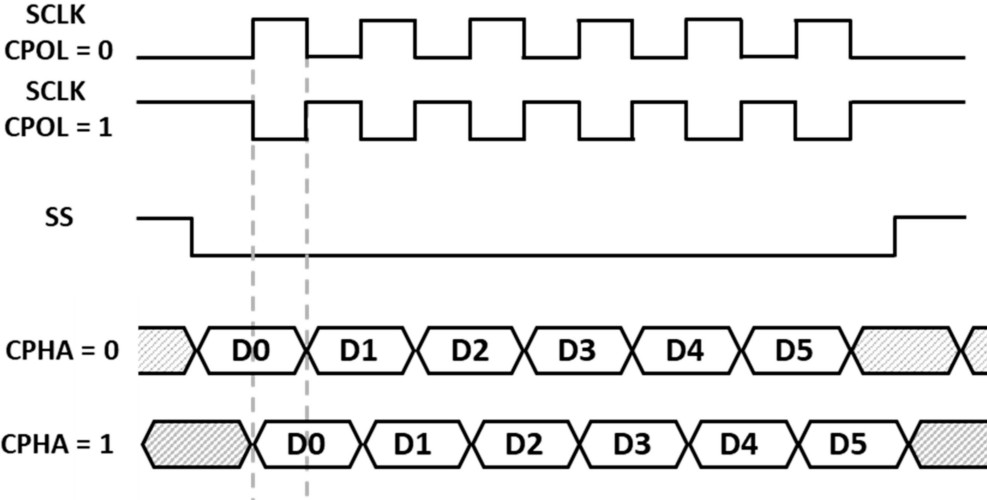

**Figure 3.** SPI clock polarity and clock phase (CPHA) working diagram.

SPI is a synchronous serial interface that can be implemented for simplex, half-duplex and full duplex. The transmitter and receiver only need to appoint a specific clock rate which is the same as the communication clock. For the circuit design, an asynchronous interface requires an additional function to handle the different clock rates on the transmitter and receiver mode. The well known design for an asynchronous interface performs a handshake-like or a first-in–first-out (FIFO) process. However, that said design may cause the chip area to increase and the cost to increase as well. In the synchronous interface, the way of exchanging data is divided into three working modes to adapt to different conditions.

Figure 4 illustrates each of the three modes of operation. Simplex communication involves a single direction for data transmission as shown in Figure 4a. Figure 4b illustrates a half-duplex mode where

the data exchange uses a bi-directional data line but does not transmit and receive simultaneously. Figure 4c shows a full-duplex mode where the exchange of data is bi-directional and can transmit and receive data simultaneously. To support a full-duplex mode, the SPI protocol is used with a faster transmission speed than the I$^2$C protocol as mentioned in [14].

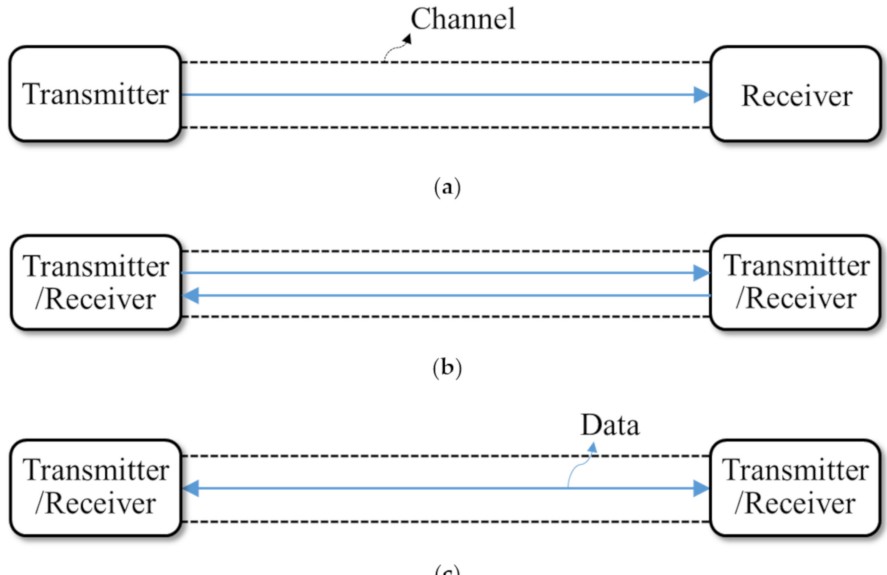

**Figure 4.** Synchronous interface working modes: (**a**) simplex mode; (**b**) half-duplex mode; and (**c**) full-duplex mode.

Considering the chip area and cost, [17] successfully realized a three-wire SPI protocol on ASIC as shown in Figure 5. The data line was named SDO for the master device to transmit data to a slave device, and SDI for the master device to receive data from a slave device. Given the sharing of data line, a mechanism is requested for making a distinction on in which direction the data line is operating and which device is it sending data to.

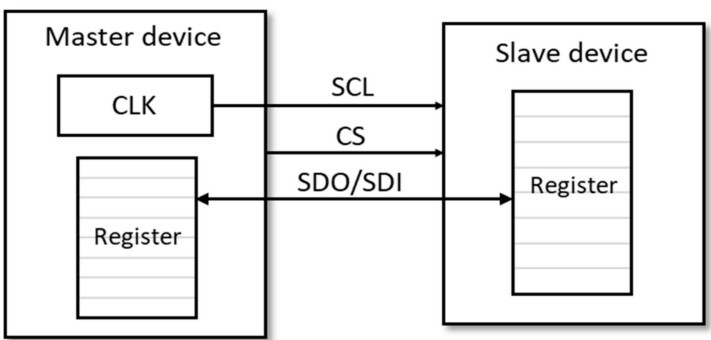

**Figure 5.** The 3-wire SPI protocol block diagram.

An instruction code was provided for deciding the direction of the data line and are listed in Table 1. There are two bits used to represent if the communication is in writing or reading mode. When the instruction code is set to '00', the SPI communication is in a writing mode and the master device will send data to the slave device. When the instruction code is set to '01', the communication mode is set to a reading mode and the master device will receive data from the slave device. In the low-cost three-wire SPI protocol design, it does not only reduce the communication lines but also removes the data synchronization circuit for an asynchronous interface.

**Table 1.** The low-cost 3-wire SPI protocol design instruction code.

| Instruction Code | Mode |
|---|---|
| 2′b00 | Write |
| 2′b01 | Read |
| 2′b10 | Don't Care |
| 2′b11 | Don't Care |

Power and cost reduction are the key important parameters to the SPI protocol market. The three-wire SPI has its limitations due to it having too many interface lines for some applications. Although the two data lines combined to one data line is an effective way to reduce the number of interface lines, in a synchronous interface, one data line cannot transmit and receive data at the same time. Therefore, the design [17] does not support the full-duplex mode for data communication and data transmission speed is limited to only one data line. Figure 6 shows the low-cost three-wire SPI protocol design realized on an FPGA and an ASIC system where FPGA is used as a master device and the ASIC as a slave device. In Figure 6a, the FPGA sends the instruction code '00' to write eight bits of data to the SRAM of the ASIC with an eight-bit address. In Figure 6b, the FPGA sends the instruction code '01' to read the SRAM data from the specified address. The master device sends the instruction code and writes the data first. After one clock pulse, the master device receives the data from the SRAM of the ASIC system. The experimental results successfully showed that the three-wire SPI protocol chip design provided an outstanding contribution for a low-cost serial transmission design.

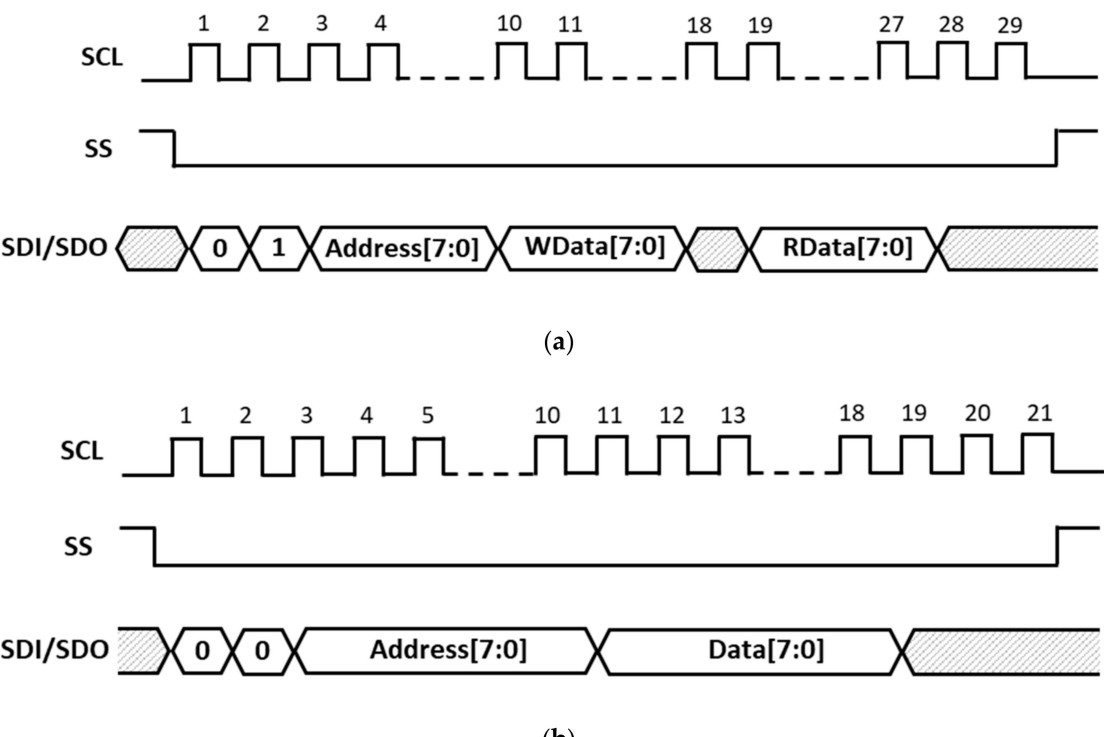

**Figure 6.** The 3-wire SPI protocol low-cost design operation waveform: (**a**) writing mode; and (**b**) reading mode.

## 3. Experimental Results and Discussion

A novel serial transmission protocol for an oscillator control interface is proposed with a lower cost and lower power consumption. In this study, a low-power synchronous preamble data line protocol presents data communication with only two interface lines which is successfully realized using the TSMC 0.18 µm CMOS process. The clock synchronous technique is proposed and verified in this study.

The low-power synchronous preamble data line protocol keeps the master device and slave device synchronized through a data line with a clock trigger. In the previous design, the data communication on a data line occurs after the clock synchronization. The low-power synchronous preamble data line protocol shares the data line function of a three-wire SPI protocol and improves the clock line. In a traditional SPI protocol, the transmission speed is limited to half of the operating frequency of the slave drive because of the sampling rate. In this design, the slave and master clocks are synchronized to the clock rate of the slave device. This enables the communication to work at full speed following the clock rate of the slave device. The design cost and power consumption were decreased through the removal of the clock line. The low-power synchronous preamble data line protocol enables an oscillator controller design that is more cost-efficient and power-efficient.

*3.1. The Low-Power Synchronous Preamble Data Line Protocol Operating Principle*

In this Section, low-power synchronous preamble data line protocol operating principle is described. Figure 7 and Table 2 shows and lists the details of the low-power synchronous preamble data line protocol operation and configuration, respectively. The lines of the low-power synchronous preamble data line protocol interface include the CS and SD lines as illustrated in Figure 7. The two-wire design eliminated the use of the SCLK signal, which is the serial clock of the three-wire SPI protocol. Moreover, the proposed two-wire interface design includes a timing synchronization mechanism to solve the clock issue and compensate for the removal of the clock line SCLK. The communication between the master and slave devices is phase synchronized. Thus, the synchronization module is designed for transmitting data with the same phase with the reference clock. When the timing synchronization is done synchronizing, the master device will transmit the data to or receive data from the slave device which is similar to the traditional SPI protocol operation.

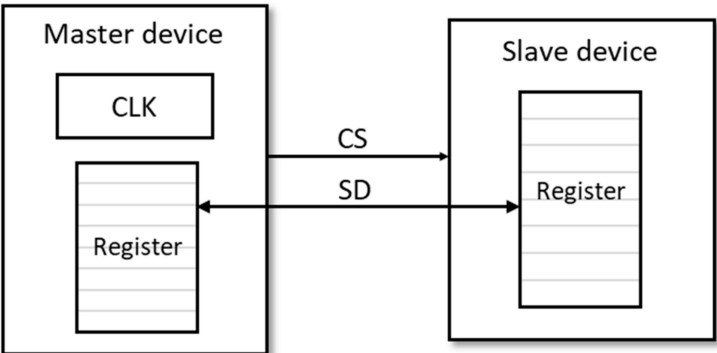

**Figure 7.** The low-power synchronous preamble data line protocol architecture.

**Table 2.** The low-power synchronous preamble data line protocol instruction code.

| Instruction Code | Mode |
|---|---|
| 2'b00 | Write |
| 2'b01 | Read |
| 2'b10 | Don't Care |
| 2'b11 | Don't Care |

In this two-wire interference design protocol, the transmission starts when the CS gets a low-level signal. First of all, the transmission is synchronized. Transmitters will send synchronized signals via the SD line. Then, the receiver detects the signal and adjusts the transmission speed to exactly the same speed as that of the transmitter. The proposed protocol then transmits the data based on the set clock rate. Because the data line is bi-directional, the transmission also needs to recognize the direction of the data through the instruction code. It determines if the device is in a transmission mode or in a receiving mode. Referring to Table 2, the two-bit instruction code sets if the data transmission line is in writing

mode or reading mode. Instruction code '00' sets the transmission of the data and '01' sets the receiving of the data. Instructions '10' and '11' are "don't cares". Figure 8 shows an example to illustrate the reading and writing modes of the proposed protocol on the memory communication system.

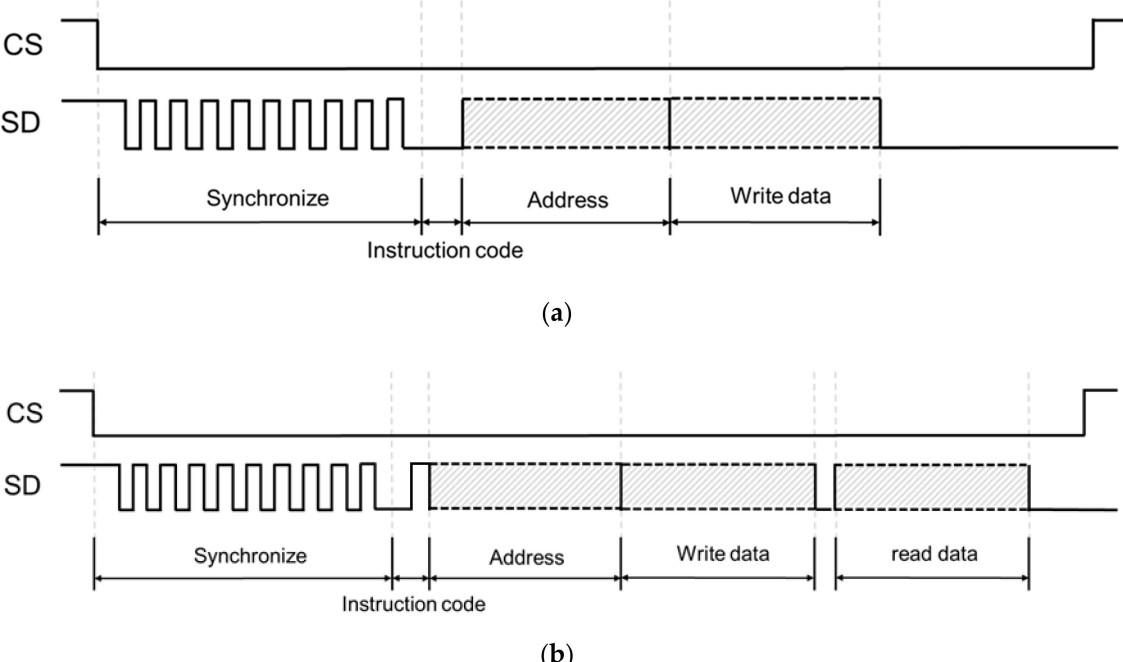

**Figure 8.** The low-power synchronous preamble data line protocol operation waveform: (**a**) the waveform of writing data, and (**b**) the waveform of reading data.

Figure 8a illustrates the operation of the writing data to the memory. First, the CS line gets a low level signal to initialize the communication. The slave device will receive the signal via the SD line according to the clock speed of the master device. The objective is to synchronize the clock of the master device and the slave device. In the synchronization process, the slave device transmits several clock cycles to the master device and the master device detects the clock rate and locks the phase. Both the master device and slave devices must be triggered in the same clock rate to ensure correct data transmission. Once synchronized, the two-bit instruction code '00' decides to write data in memory. Then, the memory addresses and data are transmitted. The sequence of the data for transmission is a two-bit instruction code, an eight-bit address, and an eight-bit datum. In Figure 8b, the waveform illustrates the operation of the proposed protocol and how to read data from the slave device SRAM. The process is similar to writing data. Synchronization makes the operating clock consistent, and the instruction code is '01' to configure the transmission to reading mode. Finally, the master device sends the memory address and waits for the return of data from the memory as instructed by the slave device.

## 3.2. The Architecture of the Low-Power Synchronous Preamble Data Line Protocol Implementation

In an oscillator control application in Figure 9, the system configuration includes one pair of master and slave devices. The master device is set as the programmer to write parameters to the memory in an oscillator chip. The master device contains a communication controller with a phase-detecting module, a phase-locking module, a writing module and a reading module. In a slave device, the oscillator provides an operating frequency as the system clock to drive the communication controller. The slave device will transmit the system clock rate to the master device for synchronization. While the clock is being synchronized, the master device writes the parameters to the memory for tuning the frequency of the oscillator chip output. In implementing a low-power synchronous preamble data line protocol verification system, the environment is inclusive of an FPGA, an ASIC, an oscilloscope, and a logic

analyzer. In the verification system, an FPGA is used as the master device while the ASIC is used as a slave device in the proposed protocol. Figure 10 illustrates the verification system. As a master device, the FPGA is flexible in supporting applications with programmable logic devices. Similar to designing a digital circuit, the FPGA function is realized by Verilog or VHDL (Very High-Speed Hardware Description Language) code. For every upload of code to the FPGA, the FPGA tools will do synthesis, placement, and routing to implement the code in a circuit. Thus, the FPGA can be reused for different applications. It only needs to change the code for every application. Having a perfect peripheral hardware is also a benefit when using an FPGA as a verification device. In this case, the instruction code and writing data are set through the buttons and switches on the FPGA board. With several switches and buttons, the master device can easily transmit different signals for verification.

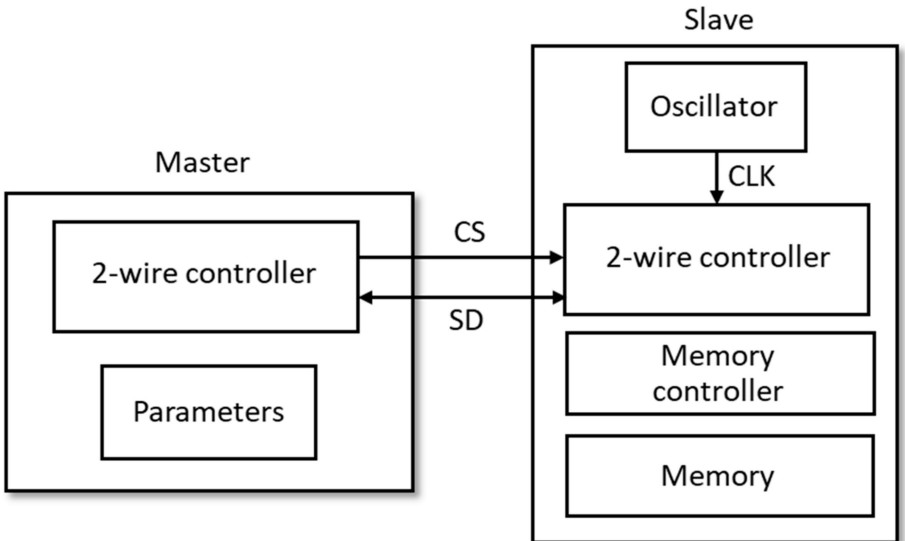

**Figure 9.** The system diagram of an oscillator control application.

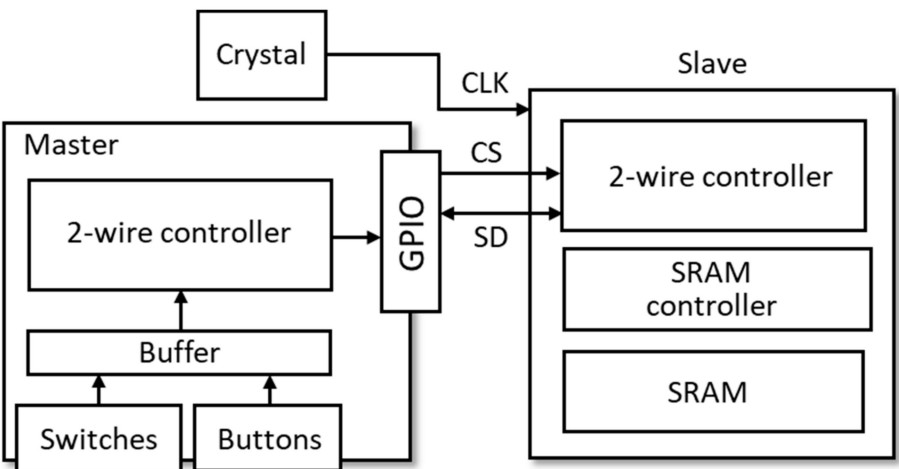

**Figure 10.** Interconnection of the master device and the slave device.

The slave device is an ASIC design that is composed of an SRAM, a memory controller, and a two-wire controller. The ASIC design was produced using the TSMC 0.18 μm CMOS process. There are three main pins used for verification, the CS, SD, and CLK pins. The CS and SD pins are connected to the general purpose input–output (GPIO) module of the FPGA. The CLK pin is provided as a system clock that is connected to a crystal oscillator. In an oscillator control application, the transmission time is short and allows flexibility in terms of transmission speed requirement. The transmission error

rarely occurs in an oscillator control application. The explanation of each part will be described in detail in the succeeding section.

### 3.2.1. Field-Programmable Gate Array (FPGA)

In a verification system, an Altera DE2-115 with Cyclone IV E FPGA is utilized as the master device. The peripheral connectors are enough for changing the test patterns and connecting it to a slave device. The Verilog code of the two-wire controller circuit is uploaded to the FPGA. Quartus II, the Altera FPGA design tool, was used to configure the peripheral hardware. For this specific FPGA model, there are eight switches, two buttons, two GPIO pins for CS and SD, and several GPIO pins for debugging.

#### Signal Configuration

The FPGA has many switches and buttons that can be used as input digital signals. To communicate with the SRAM, it needs the instruction code and address, and involves the writing of data. These signals are configured using the FPGA switches. Switches are easy to use to change the signal and verify the operation of the two-wire communication. To implement all the signal configuration, it needs 18 switches to completely describe the data structure, two switches to represent the instruction code, eight switches to define the memory address, and eight switches for writing the data. Since the switches input the signal in a parallel way, a buffer is needed to temporarily store these signals. The two-wire controller receives the clock rate from the ASIC, detects the clock rate and locks the phase. While the two-wire controller is synchronized, the signal in the buffer is transmitted to the slave device as control signals. The buttons on the FPGA are also used in this verification system. The CS line determines when the transmission begins. The button can be easily applied to the CS signal. When pressing the button, the CS line gets a low-level signal. Otherwise, it gets a high level signal. Given the peripheral components on the FPGA, it can apply any testing pattern for the low-power synchronous preamble data line protocol. As the configuration for the clock polarity and clock phase, the FPGA configures the clock polarity to '0', CPOL = 0, clock phase to '0', and CPHA = 0, as the default state. Since initially CPOL = 0, the master device clock signal is set to a low level therefore it is in an idle state. Since initially CPHA = 0, the data is acquired in the first clock edge. To keep the implementation simple, the configuration is set to the default state as CPOL = 0 and CPHA = 0 in order to clearly verify the memory operation of the low-power synchronous preamble data line protocol.

#### Device Connecting

The general purpose input/output (GPIO) is a flexible FPGA peripheral hardware. The routing to any design pin can be configured using Quartus II. By using the GPIO, digital design in FPGA can connect it with any external device by using a digital signal. The board provides 36 GPIO pins for users to configure. The CS and SD lines of the two-wire SPI controller are configured to two GPIO pins of the FPGA. Those pins are connected to the ASIC in order to control the memory. The common ground pin is supported and connected to all the ground region in the system.

### 3.2.2. Application-Specific Integrated Circuit (ASIC)

#### ASIC Design

In the proposed design, two modules were designed in ASIC. One of the modules was a two-wire controller. It sends eight clock cycles to the master device for synchronization and recognizes the instruction code to identify if it will be set in writing mode or reading mode. It also passes the memory address and data to the SRAM controller for writing data in the SRAM. For reading data, it only passes the memory address. For the clock polarity and clock phase configuration, the two-wire controller is set with CPOL = 0 and CPHA = 0, same as in the FPGA master device. Another module, the SRAM controller, is a general design for the memory control. In this case, the SRAM controller that is in

writing mode passes the data to a specific memory address while it receives the data from the SRAM when it is in a reading mode. The detailed structure is illustrated in Figure 11.

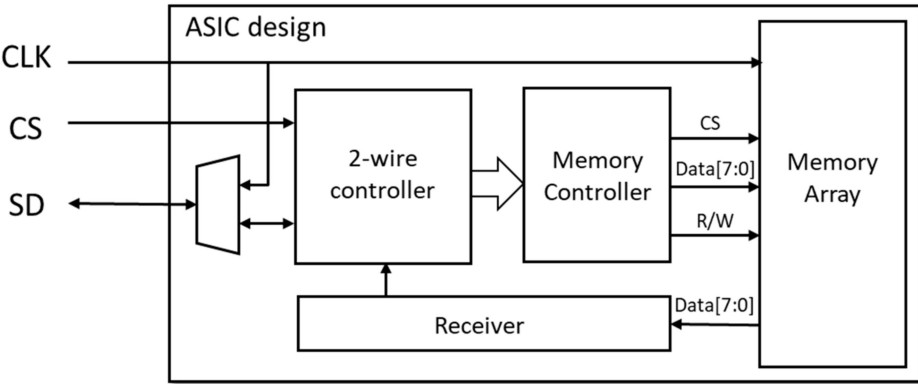

**Figure 11.** Application-specific integrated circuit (ASIC) design diagram.

Device Connecting

In a low-power synchronous preamble data line chip, there are 14 power pins and three signal pins, CS, SD, and CLK, for connecting to external devices. The 14 power pins connect to the power supply and system ground to get a stable power source. The CS and SD pins are connected to the GPIO of the FPGA for transmitting data through the low-power synchronous preamble data line protocol. The last pin, CLK, is connected to a 50 Mhz crystal and is probed to the oscilloscope for verifying the data transmission following the synchronized clock.

*3.3. Experimental Results*

A novel synchronous low-power synchronous preamble data line protocol is proposed and realized on an ASIC using the TSMC 0.18 μm CMOS process. The proposed protocol design is verified to be working correctly through a verification system which includes a built-in ASIC function embedded in a FPGA that communicates using two wires. The verification system consists of an FPGA development board, a logic analyzer, an oscilloscope, and the proposed protocol in ASIC design. The family of the FPGA board used is the Altera DE2-115 that has the Cyclone IV E core included. The low-power synchronous preamble data line protocol was implemented in an FPGA by connecting the GPIO with the ASIC and probing the oscilloscope to verify the design. The logic analyzer used is the Zeroplus LAP-C series that monitors the signal of each line between the master device (FPGA) and the slave device (ASIC). The ASIC was designed using TSMC 0.18 μm CMOS process with a 2-wire controller and an SRAM controller. The implementation of the proposed protocol is an oscillator controlling an SRAM interface.

The logic analyzer and oscilloscope illustrate the 2-wire communication. Figure 12 shows the low-power synchronous preamble data line protocol operation from the logic analyzer Zeroplus LAP-C series. In Figure 12a, it is presented that the master device reads out the initial data from the memory through the proposed protocol. The two switches on the FPGA were set to '01' to define the instruction code, and the other switches were set to '11001110' in binary as the memory address where to read the data from the memory. In the first block in Figure 12a, the master device synchronizes the clock with the slave device. After the clock is synchronized, the instruction code uses two clock cycles to configure the code '01' and set the mode to reading mode in the second block. Right after the instruction code, there is an eight-clock positive edge to send the memory address '11001110' and write the data to the slave device. Regardless of the mode, writing or reading, the data package of the low-power synchronous preamble data line the protocol memory controller sends the writing data. Thus, writing data '01010110' in the fourth block is invalid in reading mode. Once the address and data are finished transmission, the master device will receive the instruction code '01' from memory and receive the memory address

from the slave device. In the fifth block, it shows that the initial data of memory is '00000000' in binary format. This is because the memory is at its default state with a low-level voltage while the ASIC power is turned on at the beginning. After checking the initial contents of the memory, the writing data operation takes place and is presented in Figure 12b. Here, the switches were adjusted to '00' for the instruction code to set the writing data mode to memory. The address was set as '11001110' and the data '01010110' writes the eight bits to the specific memory address. In Figure 12b, the waveform of the clock synchronization happens at the first block, and the 18 bits '00_11001110_01010110' at the second, third, and fourth block will be sent to the slave device. The 18 bits of data consists of a two-bit instruction code, an eight-bit memory address, and an eight-bit datum. The verification of reading the data in the same address is shown in Figure 12c. Similar to Figure 12a, in reading the initial data, the master device sends the instruction code '01' to read the data. The memory address is the same, '11001110', to read the specific memory data. The fifth block in Figure 12c shows that the data from memory is '01010110', which is the same as the data written in the previous state shown in Figure 12b. In this experiment, the transmission error did not occur when writing or reading the indicated memory address location. The two-wire communication was successfully verified in this experiment.

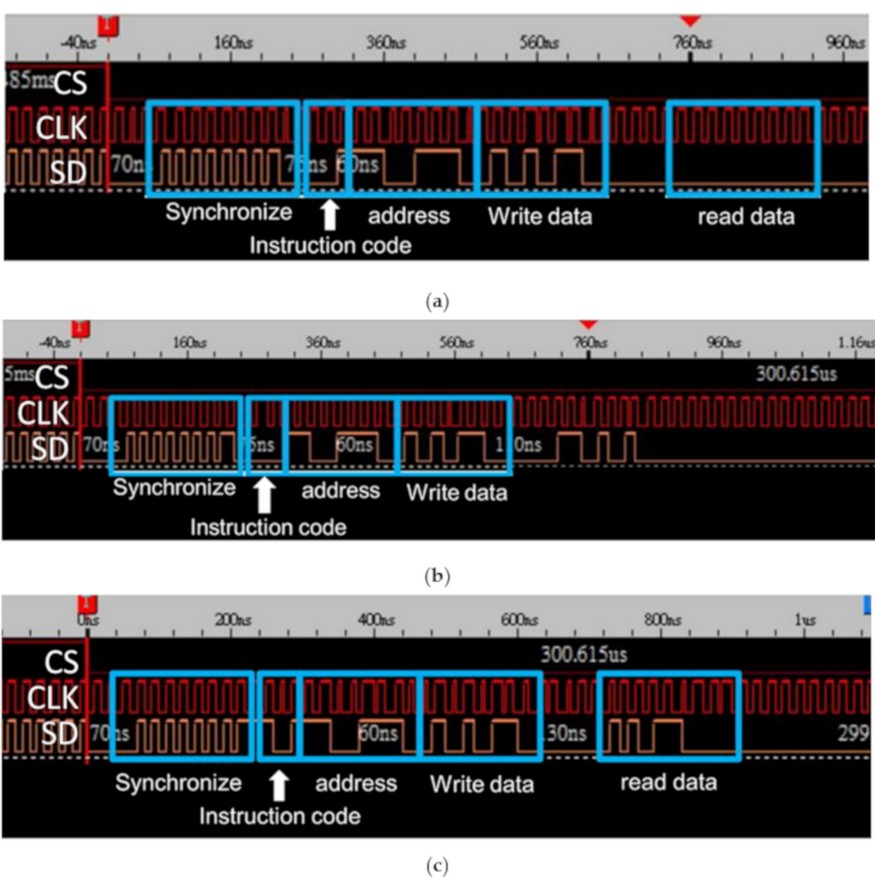

**Figure 12.** The low-power synchronous preamble data line protocol waveform in the logic analyzer: (**a**) waveform of reading the initial data from memory; (**b**) the waveform of writing the data to memory; and (**c**) the waveform of reading the updated data from memory.

The verification system setup is presented in Figure 13. The power supply provides power to the ASIC for activation. The oscilloscope probes on the clock pin of the ASIC check if it operates at a frequency of 50 MHz. The probes on the CS and SD pins on the ASIC are used to monitor the two-wire communication. The logic analyzer is accessed and is connected to the computer through the USB port where the waveform of transmission is illustrated on the Zeroplus software as shown in Figure 13.

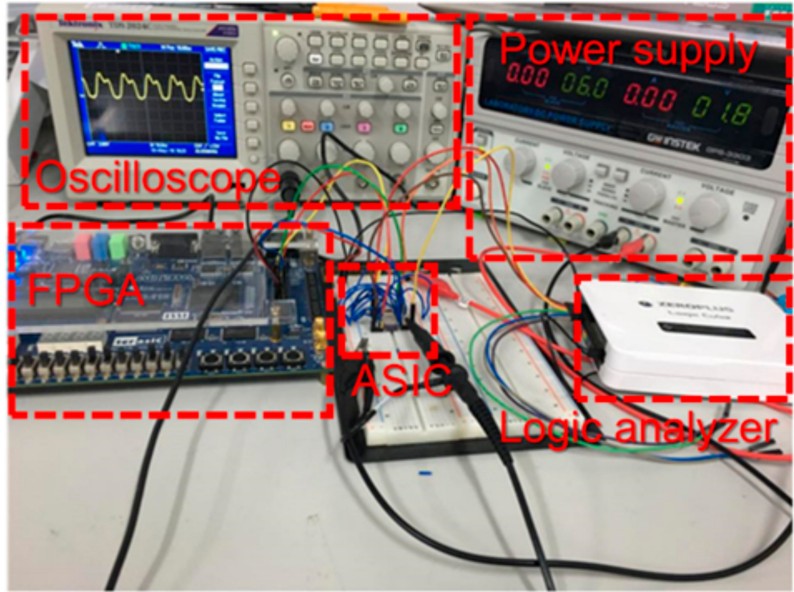

**Figure 13.** Verification environment that consists of power supply, oscilloscope, FPGA, ASIC, and logic analyzer.

The proposed design is a synchronous two-wire interface of ASIC. The ASIC consists of a two-wire controller and a memory controller for verifying the low-power synchronous preamble data line protocol by controlling the memory. The proposed protocol ASIC was fabricated using TSMC 0.18 µm CMOS process. Figure 14 shows the view of the die chip through an optical microscope.

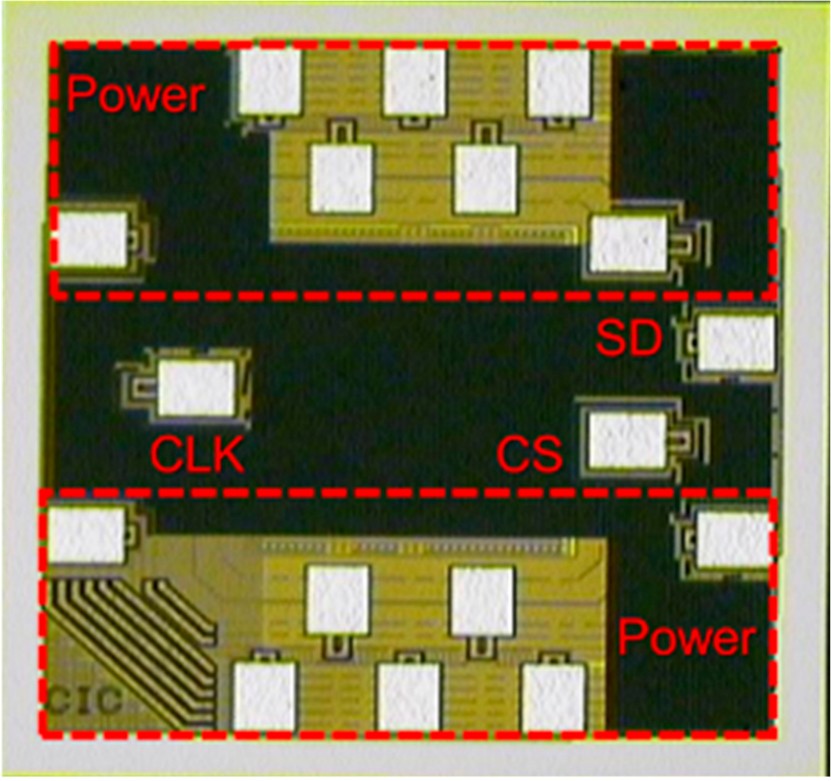

**Figure 14.** Fabricated chip view of the two-wire low-power synchronous preamble data line ASIC design through an optical microscope.

Table 3 lists the specifications of the two-wire low-power synchronous preamble data line ASIC design. The chip area is $692 \times 655$ μm$^2$ with a power consumption of 12 mW at an operating frequency of 50 MHz. The core design, two-wire controller and the SRAM controller, has a gate count of only 0.67 K gates. The SRAM controller in ASIC supports 256 addresses and eight bits of data bandwidth for communicating with the SRAM. The gate count of the SRAM is 4.4 K which comprises 86% of the whole chip. The SRAM and the proposed protocol were realized through a cost-efficient method. Table 4 lists the chip specifications and shows a side-by-side comparison between the three-wire SPI design and the proposed protocol design. At the same operating frequency of 50 MHz, the power consumption of the proposed protocol is only 12 mW, which is 36.8% lower than that of the three-wire SPI protocol design. Moreover, the chip gate count is decreased by 0.5%.

**Table 3.** Chip specification.

| Parameter | Specification | |
|---|---|---|
| Technology | TSMC 0.18 μm CMOS process | |
| Pad/core voltage | 3.3/1.8 (V) | |
| Power consumption | 12 mW | |
| Operating clock | 50 MHz | |
| Chip size (μm$^2$) | $692 \times 655$ | |
| Number of power/signal PAD | 14/3 | |
| | Memory | Full chip |
| Gate counts (K) | 4.4 | 5.07 |

**Table 4.** Comparison of the three-wire and proposed two-wire design.

| | 3-Wire SPI Design [17] | Proposed 2-Wire Design |
|---|---|---|
| Process (μm) | 0.18 | 0.18 |
| System frequency | 50 MHz | 50 MHz |
| Serial frequency | 50 MHz | N/A |
| Support configuration | Simplex, half-duplex | Simplex, half-duplex |
| Number of signal PAD | 4 | 3 |
| Power consumption | 19 mW | 12 mW ($-36.8\%$) |
| Chip size (μm$^2$) | $690 \times 655$ | $692 \times 655$ |
| Chip gate counts (K) | 5.1 | 5.07 ($-0.5\%$) |
| Memory gate counts (K) | 4.4 | 4.4 |
| Core gate counts (K) | 0.7 | 0.63 ($-10\%$) |

## 4. Conclusions

The digital controlled oscillator is a trend for a wide range of tuning frequency. The stability and precision of tuning is a great benefit for a digital controlled oscillator. In some cases, the digital controller uses a memory to store parameters for configuring different frequencies. In comparison with the traditional controlled method such as voltage control and temperature control, fine-tuning the frequency is done directly. As for digital controlling memory, it will require an additional related circuit for driving the memory. For current microelectronic devices, low cost and low power are important features to consider. Additional circuitry and low-cost are trade-off considerations. Therefore, the development of a memory controller design that is both cost-efficient and power-efficient is necessary. The low-power synchronous preamble data line protocol was proposed and verified through an FPGA and an ASIC. The ASIC design operated at a frequency of 50 MHz and communicated with the FPGA using the proposed protocol. The gate count and area of the chip are 5.07 K and $692 \times 655$ μm$^2$, respectively. Compared to the three-wire SPI protocol design in [17], the proposed protocol has a core gate count of only 0.63 K and the power consumption is 12 mW which is lower than in [17] which had 19 mW. The proposed design successfully implemented a two-wire communication with

a lower gate count and power consumption which can be used and be beneficial in the oscillator controller with its low cost and low power features.

**Author Contributions:** Conceptualization, S.-L.C. and M.-C.T.; data curation, T.-K.C. and M.-C.T.; formal analysis, T.-K.C. and M.-C.T.; funding acquisition, L.-H.W. and W.-Y.C.; investigation, M.-C.T.; methodology, S.-L.C.; project administration, C.-A.C.; resources, L.-H.W., M.-Y.L. and P.A.R.A.; software, T.-K.C.; supervision, S.-L.C.; validation, C.-A.C.; visualization, S.-L.C., M.-Y.L. and P.A.R.A.; writing—original draft, T.-K.C.; writing—review and editing, C.-A.C., W.-Y.C. and P.A.R.A. All authors have read and agreed to the published version of the manuscript.

**Funding:** This research received no external funding.

**Acknowledgments:** This work was supported by the Ministry of Science and Technology (MOST), Taiwan, under Grant numbers of MOST-108-2628-E-033-001-MY3, MOST-108-2622-E-033-012-CC2, MOST-109-2622-E-131-001-CC3, MOST-109-2221-E-131-025 and the National Chip Implementation Center, Taiwan.

**Conflicts of Interest:** The authors declare no conflict of interest.

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
