# Peer review of "A Novel Low-Power Synchronous Preamble Data Line Chip Design for Oscillator Control Interface"

_electronics, doi:10.3390/electronics9091509_

Round 1

Reviewer 1 Report

The paper proposed a low power SPI protocol using only 2 wires. The work has also been demonstrated with a fabricated chip.

- In your proposed 2-wire design, you use SD wire to sync the clock, as shown in Fig. 8. Does the clock rate at the "sync" stage in SD wire is the same as the master clock? Or it is 1/2 frequency of master clock?

- How did you realize to generate the same clock as SD "sync" in the slave? I didn't see a PLL in your Fig. 10 or any other clock replication block in your ASIC.

- After the sync, the slave clock can still drift. How do you do with it?

- How long does the "sync" stage last? And how does this "sync" stage affect the effective date transmission rate?

- After the "sync" stage, how does the salve remember the frequency and phase? Does the slave have an on-chip oscillator? Is the any way to keep slave clock always in-phase and in-freq with the Master?

- In Figure 11, please specify which signal is which.

- Please revise the section titles. In section 3, you use "Results", but it is actually about the proposed design.

- There are so many grammatical problems. Please proof read your manuscript carefully with the help of a English native speaker.

Reviewer 2 Report

Dear authors, as a first comment let me tell you that I appreciate your research and development activity. It is always interesting to see existing protocols being optimized in favor of miniaturization and cost-reduction. I'm sorry to tell you, though, that this paper is not yet in a state that justifies its publication. There are significant changes that need to be done.

I strongly suggest to choose a new name for the new protocol you are presenting. Calling it "SPI" would cause confusion, as the latter is a well-established protocol in the community with significant differences from the one presented in this paper, which for example uses a synchronization preamble on the data line at the beginning of a new transaction.

The article is very difficult to read, due to a variety of grammar errors and, in general, the use of odd syntactic constructs, for example:

  • "to realize the timing effect"
  • "is a critical week point" (instead of "weak")

Many statements are either confusing, not explained in detail or use expressions which are not part of the common technical vocabulary, for example:

  • "The timing- synchronization mechanism makes the same phase as that of the master device and slave device where the data can be transmitted by referring to the clock frequency."
  • "The marketing challenge in terms of the cost and power consumption of a 3-wire SPI design is very competitive." (the challenge is competitive?)

In some parts of the article, there seems to be a confusion between write/transmit and read/receive. For example, Fig. 8b supposedly shows a read operation, but exchanged data is labeled "write".

The article to too long:

  • It spends too much time to explain the common SPI protocol and another "3-wire SPI" protocol you proposed in an earlier publication. An explanation of these protocols belongs to the references, since in your title you aim to explain a new "2-wire SPI".
  • The time spent in describing your test setup is too long compared to the time you could have spent in explaining details of the protocol, considering that some critical ones are missing (see below).
  • There are many repetitions across the article, for example,the board used for the test, an Altera DE2-115, is described at least twice, at lines around 260 and 319.

I find it inappropriate to refer to the SPI protocol as "previous SPI protocol", as if the latter was made obsolete by your proposed standard. As a matter of fact, the SPI standard is still in wide use, and whether it will be replaced by the one you propose will be decided by the market, but this is likely to happen quite far in the future.

I did not find a detail about your protocol which I consider critical:

  • Whether the transmitters operate in push-pull or in open-collector mode, and the value of the pull-ups if in open-collector mode
  • How long the preamble should be, in your tests, to permit an acceptable phase locking at the receiver.
  • An estimate of the error rate of the protocol.

The description of the ASIC is very limited, and lacks an explanation in 3.2.2.1 on what circuit is going to use the phase information of the synchronization preamble.

An important point about the tests is that the intention is to demonstrate the 2-wire protocol, but the system clock appears to be shared between the devices (it is driven by the FPGA into the ASIC CLK pad). This effectively makes the test a 3-wire test. The authors should address this issue, showing that the protocol works in the intended conditions, that is, having two wires connecting master and slave and proving that the communication presents a good performance and reliability.

Round 2

Reviewer 1 Report

No Comments. Good work.

Author Response

Thank you so much for your positive comments. We will keep going and with high-quality papers.

Reviewer 2 Report

Dear authors,

There are visible improvements, but a few points still need care:

1) In the first review one of my comments was:

I strongly suggest to choose a new name for the new protocol you are presenting. Calling it "SPI" would cause confusion, as the latter is a well-established protocol in the community with significant differences from the one presented in this paper, which for example uses a synchronization preamble on the data line at the beginning of a new transaction

You replied the above by changing the title of the paper, but the problem I was expressing involves the text in its entirety. I reiterate the need to clarify that this protocol you are presenting is not "SPI", which refers to a specific protocol known in the community.

2) In your reply, you mention that 

This manuscript is proofread by a professional native English speaker.

This troubles me as an answer, because sentences like "However, the bus size of an SPI protocol is weak" should have been identified and corrected by a professional consultant. While there are some improvements, language is still an issue and should be resolved.

3) While you did write in the reply document for me about the absence of observed transmission errors, this information is still not mentioned in the article.

4) At line 150 and beyond, the authors make the claim that the 3-wire SPI protocol was introduced by them, but this protocol is known and widely used. As an example, 3-wire SPI is used by the Analog Devices AD9266 ADC, and many other devices.

5) Review your abstract, as it mentions 3-wire SPI which is not the main focus of this present article.

Round 3

Reviewer 2 Report

Dear authors,

Looking its evolution from the first and second versions, this third version of your article stands in much better shape, and in my opinion is acceptable for publication.

Cheers